# Role of Drying Technologies on the Drying Kinetics, Physical Quality, Aroma, and Enzymatic Activity of Pineapple Slices

Essodézam Sylvain Tiliwa [1], Isaac Duah Boateng [2], Cunshan Zhou [1] and Baoguo Xu [1,*]

1  School of Food and Biological Engineering, Jiangsu University, Zhenjiang 212013, China;
   tiliwasyl@outlook.com (E.S.T.); cunshanzhou@163.com (C.Z.)
2  Certified Group, 199 W Rhapsody Dr, San Antonio, TX 78216, USA; boatengisaacduah@gmail.com
*  Correspondence: xbg@ujs.edu.cn

**Abstract:** With a loss of about 50% of fruits and vegetables annually, there is a continuous need to improve food handling from the farm to the consumer. The solution may come partially from the selection of proper processing techniques that produce healthy and high-quality sustainable food, preserve natural resources, and contribute to prospering local economies. Pineapple is one of the most consumed fruits worldwide due to its remarkable sensorial and health-promoting attributes. Nevertheless, pineapple's high moisture content (81–86%) impedes its long-term preservation, resulting in product losses and economic, social, and environmental challenges. Drying is the oldest processing technique for most fruits and vegetables. However, the investigation of modern technologies, such as infrared drying of pineapple, is limited. Moreover, industries are investigating different methods to dry faster, thereby saving energy and reducing environmental impact. Hence, this study used four drying methods to dry pineapple slices to allow the estimation of the most promising technique: infrared drying (ID), freeze-drying (FD), convective drying (CD), and relative humidity convective drying (RHCD). The impact of these dehydration techniques on drying kinetics, physical attributes (color, texture, rehydration, microstructure), aroma, and enzymatic activity (polyphenol oxidase, peroxidase) were reported. The results showed that ID had the highest coefficient of effective moisture diffusivity and drying rates and the shortest drying period (33.45%, 36.18%, and 76.12% lower than CD, RHCD, and FD, respectively). Drying curves were successfully fitted with the parabolic and logarithmic models, which showed higher coefficients of determination and lower reduced chi-square and root mean square error than the Newton and inverse logarithmic models. FD and ID triggered minor browning indexes, leading to the brightest products. RHCD and ID slices had the highest textural values and aroma concentrations, while FD samples showed the lowest. However, FD samples had a higher rehydration ratio than other dried products and showed slight structural modifications. Regarding polyphenol oxidase and peroxidase inactivation, ID was superior, followed by CD, RHCD, and FD. The actual results suggest that infrared drying could be an efficient technique for the obtention of high-quality dehydrated pineapple fruits in a short time.

**Keywords:** pineapple; drying techniques; infrared drying; drying kinetics; product quality; physicochemical

## 1. Introduction

Currently, there exists a notable array of worldwide phenomena that are undermining the long-term viability of food and agricultural systems. The primary factor contributing to this predicament can be attributed to the escalating global population. The global population currently exceeds 7.7 billion individuals and is experiencing an annual growth rate of approximately 1.07%. Projections indicate that by the year 2050, the population will surpass 10 billion, representing an increase of over 30% compared to the present. Hence, the topic of food alternatives and its relationship with food safety and sustainability has been extensively studied. Achieving food security necessitates the harmonization of

heightened agricultural productivity and the mitigation of food demand, encompassing the reduction of food loss and waste, as well as the prevention of losses across the supply chain. A considerable amount of food is wasted at the farm level and throughout various post-farm gate activities. Approximately one-third of agricultural production designated to sustain the food supply is subject to wastage. The available data on food loss and waste indicate that fruits and vegetables constitute the primary component of food loss and waste within the food system. The reduction of food loss and waste is expected to yield positive outcomes for the promotion of sustainable production and consumption.

The pineapple fruit (*Ananas comosus*) is a characteristic fruit of tropical and subtropical regions, with the largest producers worldwide in 2019 being Costa Rica, the Philippines, and Brazil [1]. The fruit is constantly voted on the marketplace for its compelling natural aroma and taste, which balances acidity and sweetness, and is consumed either fresh or under various processed forms such as canned slices, juice, candies, jam, nectar, jelly, and dried pieces [2]. Research has documented that pineapple fruits supply numerous nutrients, taking in carbohydrates (glucose, fructose, sucrose, fibers), β-carotene, acids (malic and citric), minerals (calcium, sodium, phosphorus, magnesium, manganese, potassium, copper, zinc, iron, and chlorine), and vitamins ($B_1$, $B_2$, $B_3$, $B_6$, A, C). Additionally, it contains antioxidants and a protein digestive enzyme (bromelain from which the family name "Bromeliaceae" is derived), which contributes to relieving digestion problems and ensures the cleanness of the digestive system [3].

During the last decade, global pineapple production increased substantially to meet the evolving market demand. However, the high moisture content of pineapple (above 80% wb) can lead to postharvest losses, resulting in nutritional, economic, environmental, and social problems [4]. Therefore, preservation technologies must be applied in the early moments following the harvest.

Drying is one of the most addressed tools for extending the storage period of agricultural products. It lowers a food's water activity and prevents enzymatic reactions and microbial multiplication. Furthermore, through moisture elimination, drying can concentrate nutrients, develop new products for taste diversification, and ensure the availability of seasonal crops such as pineapple during the entire year [5].

The field of pineapple processing is primarily dominated by juice manufacturing, and the production of dried slices can offer the opportunity to add more value to the fruit and reduce the competition noticed in the juice industry. In addition, drying can reduce the volume of fruits and consequently decrease the transportation cost when conveying products to various consumer groups around the globe [6]. Throughout history, an attempt has been made to overcome the limitations of commonly used drying methods (convective drying and sun drying) in terms of low drying rate, long processing time, high energy consumption, and lower quality attributes by introducing advanced drying techniques such as microwave drying, freeze-drying, vacuum drying, and infrared drying. In this sense, there is an ongoing research trend aiming to probe the influence of several dehydration techniques on the drying kinetics and specific quality properties of agricultural products, including ginger [7], *Ginkgo biloba* seeds [8], and black carrot pomace [9].

These precedent studies exhibited that different drying methods have diverse patterns in end-product moisture elimination and physicochemical properties. Thus, selecting the appropriate dehydration technique for each agricultural product, considering the processing duration and the quality of the final product, is of stupendous importance. In the current work, we propose to investigate the action of infrared drying, convective drying, relative humidity convective drying, and freeze-drying on the dehydration of pineapple slices. In convective drying, one of the most important parameters is the humidity of the air circulating in the drying chamber. Proper control of the humidity in the dryer can improve the drying efficiency by keeping a suitable vapor pressure gradient in the sample. In a previous work [7], the control of the humidity during convective drying could improve drying kinetics and the antioxidant profile of ginger. Freeze-drying eliminates moisture at lower temperatures by sublimation, thus leading to the excellent quality of final products

due to the ability of low temperatures to slow down chemical and biological reactions. Nevertheless, it involves longer dehydration periods and consumes more energy [10]. Infrared drying is gaining a reputation as the optimum drying method for many agricultural products. Infrared radiations impose molecular vibrations on the dried sample and ensure its rapid and homogeneous heating during drying. It reduces energy consumption, drying time, and provokes the higher retention of nutrients [11].

Compared to nutritional attributes, the most noticeable characteristics of processed pineapples are color, aroma, texture, and taste [12]. These properties are the most perceived by consumers; thus, ensuring their preservation or improvement during processing will enhance end-product marketability. It is also well established that changes in color and texture result from oxidations and structural modifications, respectively. Therefore, assessing the activity of enzymes, the browning index, the rehydration, and the microstructure is also important.

Izli et al. [10] and Malaikritsanachalee et al. [13] evaluated the drying kinetics and color of pineapples during microwave drying, convective drying, and freeze-drying on one hand and hot air drying on the other hand. Similarly, Ponkham et al. [14] assessed the kinetics of moisture elimination and the color and shrinkage of pineapple pieces during convective drying combined with far-infrared radiation. Boateng et al. [8] studied the influence of hot-air drying, freeze-drying, infrared drying, and pulsed-vacuum drying on the quality, bioactive elements, antioxidant activity, and toxic compounds of *Gingko biloba* L. seeds. In addition to convective drying and freeze-drying, Polat et al. [9] also investigated the role of conductive hydro drying, microwave drying, and vacuum convective drying on the volatiles, color, and polyphenols of black carrot pomace.

However, to date, studies on the extensive analysis and comparison of the action of several drying techniques on the drying kinetics, physical quality, aroma, and enzymatic activities of pineapple fruits are scarce. In addition, the investigation of modern technologies, such as the infrared drying of pineapple, is very limited.

Therefore, this paper intends to elucidate the effect of convective drying (CD), relative humidity convective drying (RHCD), infrared drying (ID), and freeze-drying (FD) on pineapple dehydration kinetics, physical quality, final product quality (color, texture, rehydration, and microstructure), the aroma profile, and the enzymatic activities (polyphenol oxidase and peroxidase). From this perspective, the study will provide tangible directions for choosing the most effective drying method based on time fresaving and quality preservation.

## 2. Material and Methods

### 2.1. The Pineapple Fruits and Chemicals

Fully ripe pineapples (var. MD2) with a similar size used in this work originated from a local market in Zhenjiang (P.R of China). Fruits were refrigerated at 4 °C for conservation and dried in the first week following their acquisition. Before drying experiments, pineapple fruits were washed, and the pulp was obtained as a cylinder of 30 mm internal diameter and 80 mm external diameter, using a stainless-steel pineapple peeling and coring tool; a stainless-steel knife was used to obtain slices of $5 \pm 1$ mm thickness. Tissue papers were used to remove excess moisture on the surface of slices. The initial moisture content of fresh pulp ($83.71 \pm 0.44\%$ wet basis) was measured at 105 °C in a traditional lab oven until a steady weight was reached. The initial total soluble solids using a pocket refractometer (ATAGO CO., LTD, PAL-1, Tokyo, Japan) was $14.89 \pm 0.70$ °Brix, while the acidity (pH) was $3.20 \pm 0.01$ (PHS-3G, INESA, Shanghai, China). To ensure the homogeneity of results, only the middle of the fruits was used during the study [15].

Sodium phosphate (monobasic and dibasic), polyvinylpolypyrrolidone (PVPP), hydrogen peroxide, and Triton X-100 were acquired from Sinopharm Chemical Reagent Co., Ltd. (Shanghai, China); guaiacol and catechol originated from Sigma-Aldrich Co., Ltd. (St. Louis, MO, USA).

## 2.2. Drying Procedures

The experimental flowchart is presented in Figure 1. Drying experiments of pineapple slices were carried out employing convective drying, relative humidity convective drying, infrared drying, and freeze-drying until the ultimate moisture content was below 0.2 g water/g dry solid. A total of 300 g of slices (mass of 12 slices) was weighed, and the weight reduction of 4 slices was measured in triplicate during drying using a digital balance (BSA2202S, Sartorius AG, Göttingen, Germany). Drying chambers were heated for 45 min to reach uniform experimental conditions in all thermal processes. After the drying, dried products were cooled at room temperature, sealed in plastic bags, and kept in a desiccator for other analyses. Each drying experiment was performed in triplicate.

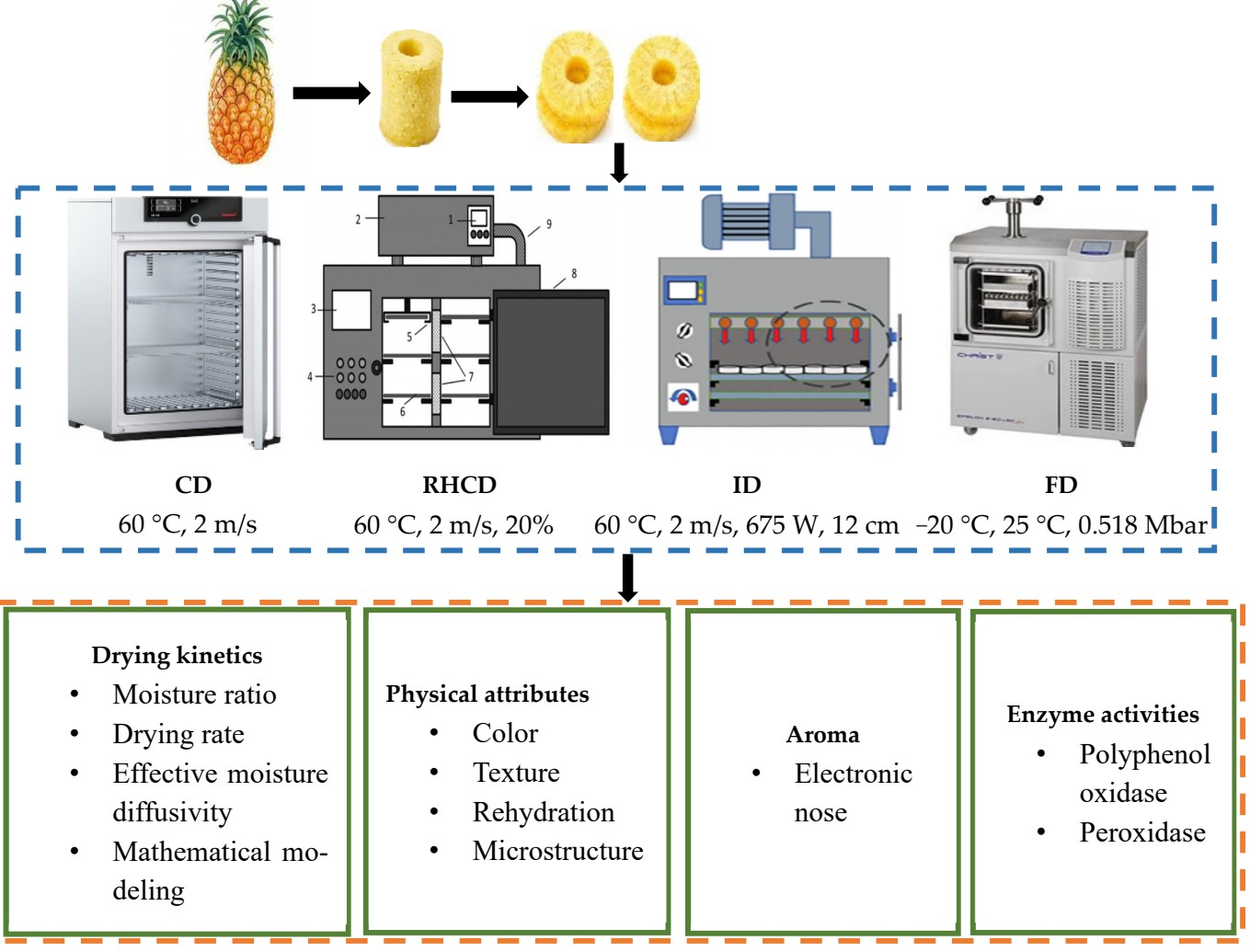

**Figure 1.** Experimental flowchart. Note: CD: convective drying, RHCD: relative humidity convective drying, ID: infrared drying, FD: freeze-drying. Note: (1) Water bath control panel. (2) Water bath. (3) Dry parameter display. (4) Dry parameter switch. (5) Weighing plate. (6) Loading tray. (7) Circulating air fan. (8) Dryer switch door. (9) Transport pipe.

### 2.2.1. Convective Drying

Fresh pineapple slices were dehydrated in an electric convective dryer (Memmert GmbH + Co. KG, Bavaria, Germany) to avoid condition fluctuations after heating the drying chamber. Samples were displayed on the dryer trays, and the dehydration process was effectuated at processing conditions of 60 °C and an air circulation of 2 m s$^{-1}$.

### 2.2.2. Relative Humidity Convective Drying

In RHCD, the temperature and the airflow parameters were identical to those set during the convective drying (60 °C and 2 m s$^{-1}$). Nevertheless, the relative humidity was automatically controlled at 20% during drying.

### 2.2.3. Infrared Drying

In infrared drying, there are four principal parameters to consider: temperature, infrared power, air velocity, and distance between infrared tubes and fresh foods. ID was carried out in an intermediate-wave infrared dryer (Samkoon Co. Ltd., Shenzhen, China) at 60 °C and 2 m s$^{-1}$ using 3 radiation sources of 225 W each. The distance separating the infrared lamps and samples was 12 cm. A comprehensive description of this infrared equipment can be found in the published paper by Boateng and Yang [8].

### 2.2.4. Freeze-Drying

Pineapple slices were preliminarily frozen at −20 °C a day before the FD. Afterward, samples were dried in a freeze-dryer (Epsilon 2-6D LSCplus, Martin Christ Ltd., Osterode am Harz, Germany) at 25 °C under 0.518 Mbar vacuum pressure. The cold trap in the equipment was set at −95 °C.

### *2.3. Drying Kinetics*
### 2.3.1. Moisture Ratio and Drying Rate

The kinetics of moisture ratio (MR) and drying rate (DR) during CD, RHCD, ID, and FD are below. For this purpose, samples (3 sets of 4 slices) were each weighed 15 min during CD, RHCD, and ID and 2 h during FD until the desired moisture content was attained using an analytical lab balance (BSA2202S, Sartorius AG, Göttingen, Germany). The moisture ratio was deduced following Equation (1) [13]:

$$MR = \frac{M_t - M_e}{M_0 - M_e} \tag{1}$$

where $M_0$, $M_e$, and $M_t$ are the moisture content (db) of the sample (g/g) in the beginning, at the equilibrium, and at a random time $t$ (h) during drying, respectively. Because the value of $Me$ is minor compared to $M_0$ and $M_t$, the above-mentioned equation can be simplified (Equation (2)) as follows [16]:

$$MR = \frac{M_t}{M_0} \tag{2}$$

The subsequent equation (Equation (3)) was used to compute the drying rate during the dehydration of pineapples [7]:

$$DR = \frac{M_{t1} - M_{t2}}{t_2 - t_1} \tag{3}$$

This equation denotes the variation of the moisture content (g/g) expressed as $(M_{t1} - M_{t2})$ in between respective drying times $t_1$ and $t_2$ (h).

### 2.3.2. Effective Moisture Diffusivity

The effective moisture diffusivity ($D_{eff}$) during various drying methods was evaluated through Fick's second law of diffusion. Several assumptions were made, taking in steady diffusivity coefficients and temperature, minor external resistance to moisture transfer, uniform moisture content, and the food being unidimensional with no shrinkage during drying. Equation (4) is in line with these hypotheses [17]:

$$MR = \frac{m_t - m_e}{m_i - m_e} = \frac{8}{\pi^2} \exp\left(-\frac{\pi^2 D_{eff} t}{4L^2}\right) \tag{4}$$

The $D_{eff}$ was computed using the equation of the slope of the curve Ln *(MR)* against the drying time according to Equation (5) [17]:

$$Slope = -\frac{\pi^2 D_{eff}}{4L^2} \tag{5}$$

MR is the moisture ratio; $m_i$, $m_t$, and $m_e$, represent the moisture content on a dry basis (g/g) at the beginning, at a random time *t* (s) during drying, and at the equilibrium, respectively; *L*, is the half-thickness of pineapple slices in meter.

### 2.3.3. Mathematic Modeling

Four thin-layer mathematical models were used for fitting the moisture ratio curves after preliminary screening in OriginPro9.8 software. These comprise the Newton model (Equation (6)), the logarithmic model (Equation (7)), the parabolic model (Equation (8)), and the inverse-logarithmic model (Equation (9)) [18]:

$$MR = \exp(-kt) \tag{6}$$

$$MR = a\exp(-kt) + c \tag{7}$$

$$MR = a + bt + ct^2 \tag{8}$$

$$MR = a - b \times \ln(t + c) \tag{9}$$

*a*, *b*, *c*, and *k* are empirical constants, while *t* is the dehydration time.

The best statistical model is the one with the smallest root mean square error (RMSE) and chi-square ($\chi^2$) and the highest coefficient of determination ($R^2$). The mathematical expressions (Equations (10)–(13)) of these statistical parameters are given below [7]:

$$RMSE = \sqrt{\frac{\sum_{i=1}^{N}\left(MR_{pre,i} - MR_{\exp,i}\right)^2}{N}} \tag{10}$$

$$\chi^2 = \frac{\sum_{i=1}^{N}\left(MR_{\exp,i} - MR_{pre,i}\right)^2}{N - n} \tag{11}$$

$$R^2 = \frac{\sum_{i=1}^{N}\left(MR_{\exp,i} - MR_{pre,i}\right)^2}{\sum_{i=1}^{N}\left(\overline{MR}_{\exp,i} - MR_{pre,i}\right)^2} \tag{12}$$

$$\overline{MR}_{\exp,i} = \frac{\sum_{i=1}^{N} MR_{\exp,i}}{N} \tag{13}$$

N is the number of observations, *n* is that of constants, $MR_{\exp}$ is the moisture ratio during experiments, and $MR_{pre}$ is the predicted moisture ratio.

### 2.4. Physical Quality

2.4.1. Color

A colorimeter (Minolta CR-400, Konica Minolta, Tokyo, Japan) was used to record color parameters, namely the brightness *(L\*)*, the redness coordinate *(a\*)*, and the yellowness coordinate *(b\*)*. From these attributes, the color change ($\Delta E$) between dried samples and the fresh material and the browning index *(BI)* were derived following Equation (14) and Equation (15), respectively [8,19]:

$$\Delta E = \sqrt{(L^* - L_0)^2 + (a^* - a_0)^2 + (b^* - b_0)^2} \tag{14}$$

$$BI = 100 \times \left( \frac{X - 0.31}{0.17} \right) \tag{15}$$

where

$$X = \frac{(a^* + 1.75L^*)}{(6.645L^* + a^* - 3.012b^*)} \tag{16}$$

The colorimeter was initially calibrated on a white plate (Y = 85.6, x = 0.3162, y = 0.3238), and all measurements were replicated 6 times.

### 2.4.2. Texture

The texture profile test was conducted using a cylindrical probe (TAXT Plus, Stable Micro Systems Ltd., Godalming, UK) as depicted by Boateng, Yang, et al. [17] with minor modifications. Three slices from each sample group were double-compressed with the probe until breakage, and the experiment was replicated 3 times. The compression strain was 40%, and the trigger force was 5 g. The probe pre-test, test, and post-test speeds were 3 mm/s, 2 mm/s, and 3 mm/s, respectively. The interval between two consecutive compressions was 5 s, and the target distance was 5 mm. Six texture parameters were recorded, specifically the hardness, the springiness, the cohesiveness, the gumminess, the chewiness, and the resilience. Only maximum values were retained, and the average was reported.

### 2.4.3. Rehydration Ratio (RR)

Dried pineapple slices in an amount of 2.5 g were immersed in 25 °C distilled water for 30 min. Such a time was chosen to understand the behavior of pineapple snacks if they were to enter a breakfast preparation. Soaked slices were moved out of the water every 5 min, blotted with tissue paper, and weighed. Afterward, slices were immediately returned to the liquid to continue the process. Rehydration experiments were performed in triplicate for each sample group. RR was determined with Equation (17) [20]:

$$RR = \frac{M}{M_0} \tag{17}$$

where $M_0$ is the mass of the dried sample, and $M$ is the mass of the rehydrated slice.

### 2.4.4. Microstructure

The microscopic structures of dehydrated samples were performed in triplicate through a scanning electron microscope (S-3400 N, Hitachi Ltd., Tokyo, Japan). First, sectioned samples were displayed on copper adhesives and covered with a slim gold layer. Then, microscopic images were obtained at an accelerating voltage of 15 kV and magnified at ×100.

### *2.5. Aroma*

The aroma profile of dried pineapples was acquired employing an electronic nose device (PEN3.5, AIRSENSE Analytics GmbH, Schwerin, Germany), which combines 10 specific aromatic sensors. An amount of 2 g of pineapple slices cut into fine pieces was collected in 20 mL glass tubes in triplicate for each sample group and incubated for 30 min in a distilled water bath at 40 °C. For e-nose analysis, the system was first cleaned by flushing air for 180 s, followed by a detection time of 120 s with an invariable injection flow rate of 400 mL/min. Information about the sensors composing the electronic nose device and the species they can detect is provided in Table 1.

### *2.6. Enzymatic Activities*

The activity of polyphenol oxidase (PPO) and that of peroxidase (POD) was assayed in triplicate according to the protocol of Alolga et al. [17] with minor modifications. An amount of 2 g of samples was first introduced in an enzyme extraction solution (10 mL)

previously obtained by mixing equal volumes of 0.2 M sodium phosphate buffer (pH 6.6), 2% (*w/v*) of polyvinyl polypyrrolidone, and triton X-100 (1% *v/v*). The solution containing the sample was homogenized with a magnetic stirrer (400 rpm) for 1 h for proper extraction, centrifuged for 30 min at (10,000 rpm, 4 °C), and the supernatant was collected for enzymatic analyses.

**Table 1.** Electronic nose sensors and their characteristics.

| Sensor Number | Official Name | Detected Species |
|:---:|:---:|:---:|
| S1 | W1C | Aromatic organic elements |
| S2 | W5S | Nitrogen oxides |
| S3 | W3C | Ammonia and aromatic components |
| S4 | W6S | Hydrogen gas |
| S5 | W5C | Aromatic alkanes and nonpolar organic compounds |
| S6 | W1S | Methane and methyl groups |
| S7 | W1W | Organic sulfides |
| S8 | W2S | Alcohols, aldehydes, and ketones |
| S9 | W2W | Aromatic elements, inorganic sulfides, and organic chemicals |
| S10 | W3S | Methane and aliphatic compounds |

For PPO activity, 50 µL of pineapple extract was reacted with 0.1 mL of 0.1 M catechol solution and 1.95 mL of 0.2 M sodium phosphate buffer (pH 6.6). The mixture was vortexed, and the absorbance was read at 410 nm using a UV–VIS spectrophotometer (Shanghai PGENERAL Ltd., Shanghai, China).

POD analysis was carried out by homogenizing 40 µL of the extract with 150 µL of guaiacol (4% *v/v*), 150 µL of 1.5% $H_2O_2$ (*v/v*), and 2.66 mL of sodium phosphate buffer (pH.6.6). The absorbance value was obtained at 470 nm. In both cases, the blank solution was obtained by substituting the sample extract with the same volume of distilled water. The enzymatic activities (EA) were calculated using Equation (18) [8]:

$$EA, \left( U \cdot g^{-1} \right) = \frac{\Delta A}{0.01 WT} \tag{18}$$

$\Delta A$ is the variation of the absorbance during the reaction time *T*; *W* is the weight in g of the pineapple sample.

### 2.7. Statistical Analysis

All experimental procedures were carried out in triplicates. Results were given as average ± standard deviation, and the test of variance (one-way ANOVA) was performed employing the software SPSS 26.0 (SPSS Inc., Chicago, IL, USA) at the significance level of $p < 0.05$ using Tukey's test. OriginPro9.8 software version 2021 (OriginLab Corporation, Northampton, MA, USA) was used for plotting graphs.

## 3. Results and Discussion

### 3.1. Drying Kinetics

Figure 2 shows the drying kinetic curves of pineapple under different drying situations. Drying times were determined to be 5.83 h, 6.08 h, 3.88 h, and 16.25 h, respectively, for CD, RHCD, ID, and FD, according to Figure 2A. These observations suggest that ID reduced drying time by 33.45%, 36.18%, and 76.12% compared to CD, RHCD, and FD, respectively.

Fresh pineapple slices are very porous and contain much water, mainly in a free state. Therefore, technologies that can increase the cellular vapor pressure will result in enhanced moisture expelling. The lower drying time in ID can be ascribed to the impact of infrared radiations, which quickly increased the temperature at the core of the sample and ensured a homogeneous heat transfer and rapid evaporation [11,21]. In addition, the high

temperature in CD, RHCD, and ID was more efficient in eliminating moisture than the low temperature in FD. This is in line with the findings of Izli et al. [10] when drying pineapple using freeze-drying, convective drying, and microwave drying and Osae et al. [7] when drying ginger using FD, ID, RHCD, and microwave drying. This observation is proven by Figure 2B, which presents the kinetics of moisture ratio as a function of time. Thermal treatments had sharper curves in comparison to FD, with IR being superior, followed by RHCD and CD. However, the CD process outperformed RHCD around the end of the drying period, which can be explained by the increased drying rate. The moisture ratio curves presented no constant rate and thus implied that the moisture was chiefly eliminated through diffusion. Similar observations were noticed for sour cherries [22].

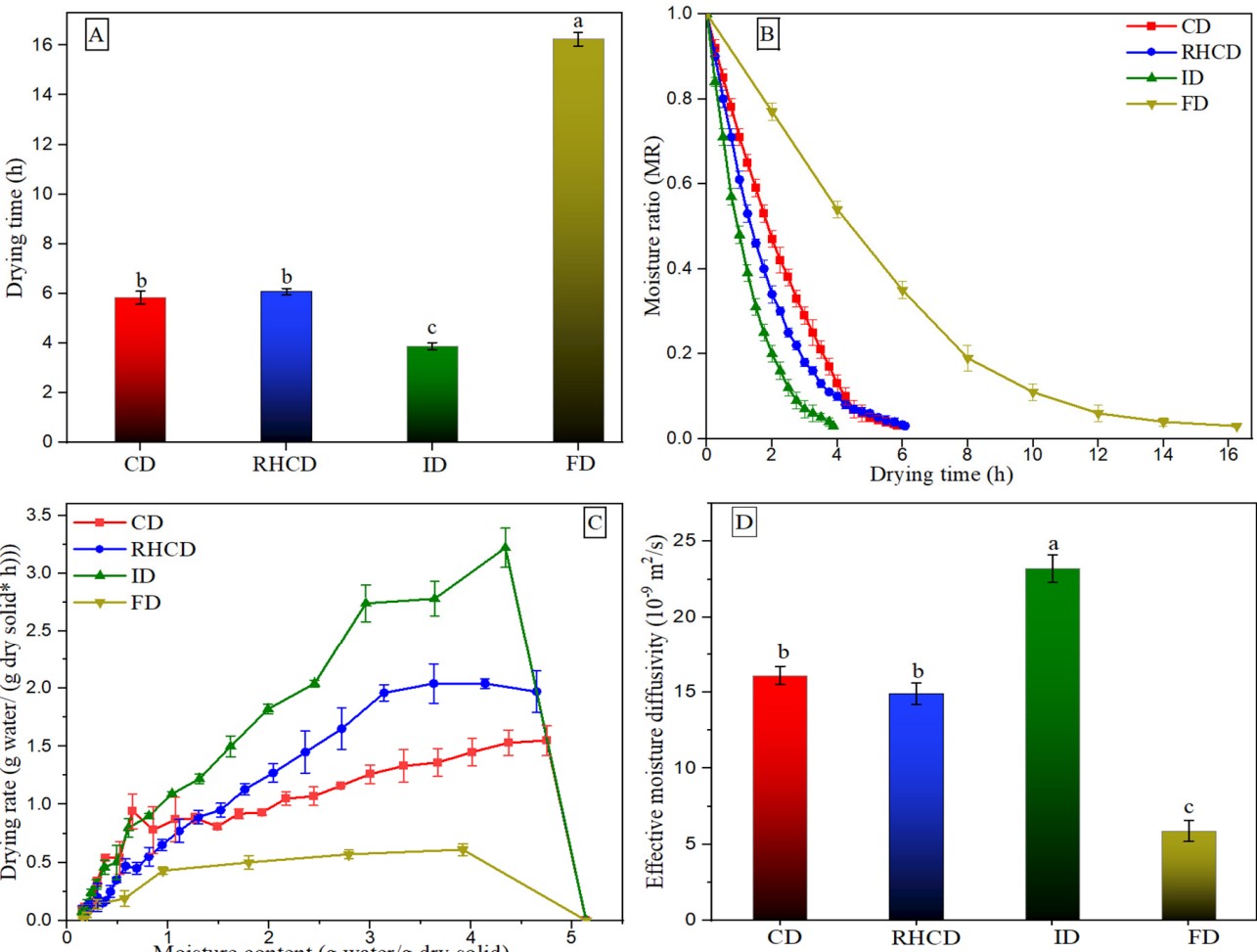

**Figure 2.** Effect of different drying methods on the drying kinetics and effective moisture diffusivity of pineapple. (**A**) Effect of different drying methods on the total drying time of pineapple. (**B**) Moisture ratio curves under different drying methods. (**C**) Effect of different drying techniques on the variation of the drying rate of pineapple. (**D**) Effective moisture diffusivity of pineapple during drying. Note: Different letters on data bars show significant differences ($p < 0.05$). CD: convective drying, RHCD: relative humidity convective drying, ID: infrared drying, FD: freeze-drying.

The curves of various drying rates plotted against the moisture content on a dry basis are presented in Figure 2C. As the moisture content progressively decreased, the drying rate was also reduced. The falling rate governed the dehydration process globally, with some exceptions. In RHCD, the drying rate increased at the beginning of the drying process up to the reach of the moisture content of 3.63 g water/g dry solid before noticing a continuous reduction. However, the drying rate of CD was enhanced when the moisture content was below 1.5 g water/g dry solid. This observation can be associated with attaining

a maximum temperature at the core of the sample around that period. Similarly to the observation in MR, the drying rate was the highest in ID, while FD recorded the lowest. Therefore, infrared drying is an ideal technique that can be utilized to profoundly speed up the dehydration of pineapple and hence contribute to saving energy and time [5]. The drying rate in the early periods of all drying methods was enhanced because a large amount of moisture in a free state was easily lost, and the phenomenon was more visible in heat-based operations. However, the drying rate dramatically decreased at a very low moisture content. It can be correlated to the interaction between nutrients, especially sugars, with water molecules, making moisture release difficult. This is in accord with the findings of Chin et al. [23] during the dehydration of kiwifruits.

The moisture diffusivity coefficient ($D_{eff}$) was calculated to further apprehend the patterns of moisture movement in pineapple slices during drying (Figure 2D). The values of $D_{eff}$ varied from $5.89 \times 10^{-9}$ m$^2$/s to $2.32 \times 10^{-8}$ m$^2$/s with FD and ID having the lowest and the highest values, respectively. $D_{eff}$ values well supported what was explained above and might be the reason for the high drying rate and short drying time reported for ID. Moreover, there was no significant ($p > 0.05$) difference in moisture diffusivity between CD and RHCD. In this sense, heat is more ideal than lower temperatures to enhance moisture movement from the inner samples toward their surfaces and improve evaporation when moisture is exposed to hot air. In addition, ID, due to its high heating ability, obtained a superior $D_{eff}$ value compared to CD and RHCD. Overall, we can deduce that the higher the $D_{eff}$, the faster the dehydration process [20]. The ID could be an ideal alternative to conventional hot-air drying regarding drying kinetics.

Different drying processes of pineapple slices were modeled using four thin-layer mathematical models. The statistical parameters and the empirical constants of each model are presented in Table 2. Overall, all the models used (Figure 3) can explain the drying process of pineapple as the $R^2$ ranged from 0.927 to 0.998.

**Table 2.** Modeling of pineapple dehydration curves: model constants and parameters.

| Mathematical Models | Model Constants and Parameters | Drying Methods | | | |
|---|---|---|---|---|---|
| | | CD | RHCD | ID | FD |
| Newton | $k$ | 0.501 | 0.561 | 0.829 | 0.209 |
| | $R^2$ | 0.944 | 0.995 | 0.993 | 0.972 |
| | RMSE | 0.120 | 0.031 | 0.041 | 0.089 |
| | $\chi^2$ ($10^{-4}$) | 144.500 | 9000.774 | 16.400 | 7.900 |
| Logarithmic | $a$ | 1.183 | 1.057 | 1.054 | 1.104 |
| | $k$ | 0.375 | 0.569 | 0.777 | 0.195 |
| | $c$ | −0.117 | −0.004 | −0.023 | −0.028 |
| | $R^2$ | 0.987 | 0.998 | 0.998 | 0.980 |
| | RMSE | 0.057 | 0.022 | 0.023 | 0.076 |
| | $\chi^2$ ($10^{-4}$) | 32.700 | 4.983 | 5.165 | 57.600 |
| Parabolic | $a$ | 1.023 | 0.898 | 0.922 | 0.988 |
| | $b$ | −0.335 | −0.319 | −0.491 | −0.132 |
| | $c$ | 0.028 | 0.029 | 0.068 | 0.005 |
| | $R^2$ | 0.997 | 0.982 | 0.991 | 0.991 |
| | RMSE | 0.026 | 0.060 | 0.046 | 0.049 |
| | $\chi^2$ ($10^{-4}$) | 6.709 | 35.900 | 20.700 | 23.900 |
| Inverse logarithmic | $a$ | 0.920 | 0.508 | 0.412 | 0.669 |
| | $b$ | 0.488 | 0.277 | 0.295 | 0.245 |
| | $c$ | 0.713 | −0.066 | −0.051 | −1.398 |
| | $R^2$ | 0.972 | 0.961 | 0.974 | 0.927 |
| | RMSE | 0.086 | 0.082 | 0.068 | 0.122 |
| | $\chi^2$ ($10^{-4}$) | 73.700 | 67.800 | 47.300 | 147.800 |

Note: $a$, $b$, $c$, and $k$ are the empirical constants of different models; $\chi^2$ is the reduced chi-square; $R^2$ is the coefficient of determination; RSME is the root mean square error; CD: convective drying, RHCD: relative humidity convective drying, ID: infrared drying, FD: freeze-drying.

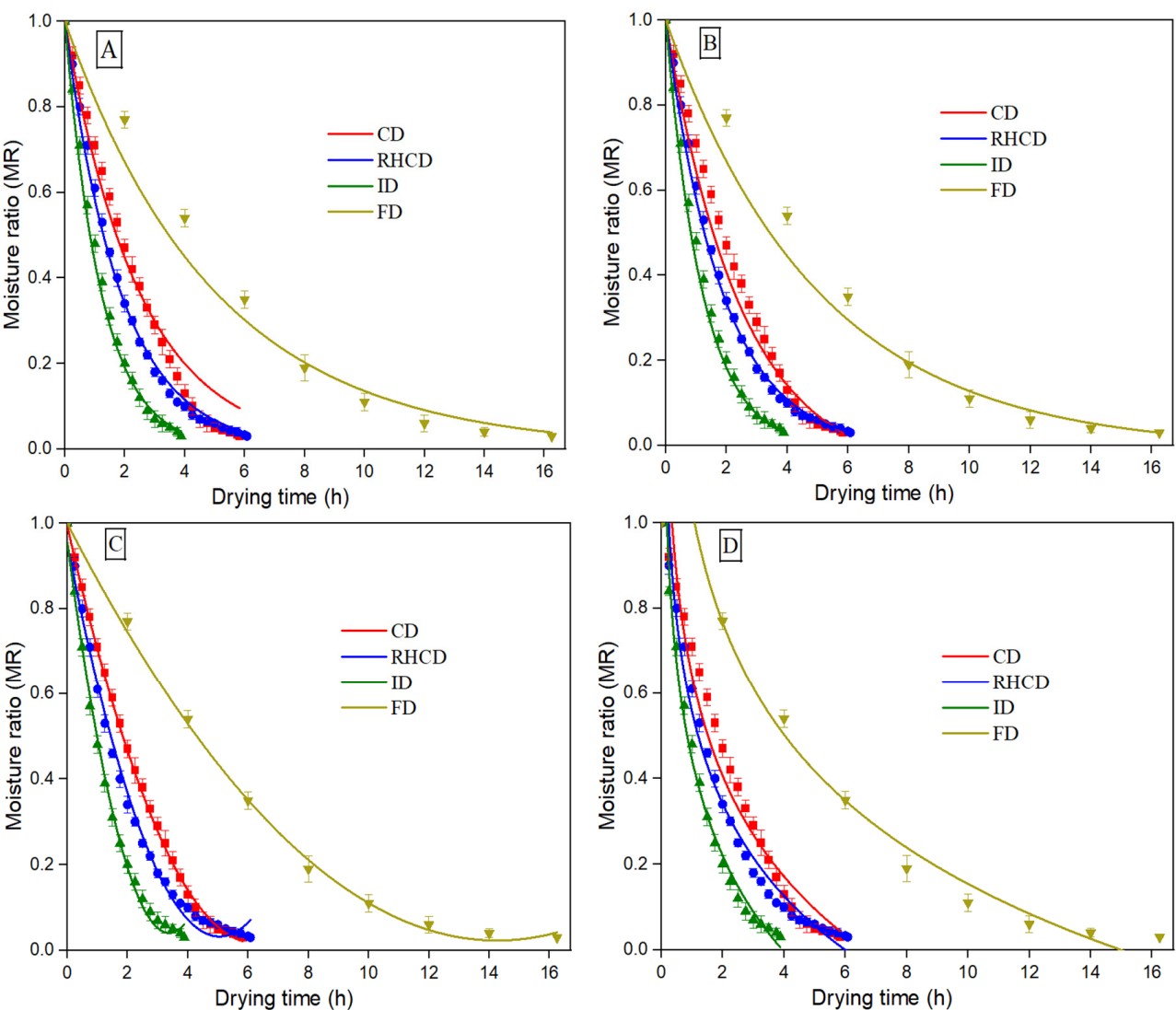

**Figure 3.** Modeling of the moisture ratio curves of pineapple slices. (**A**) Newton model, (**B**) logarithmic model, (**C**) parabolic model, (**D**) inverse-logarithmic model. Note: CD: convective drying, RHCD: relative humidity convective drying, ID: infrared drying, FD: freeze-drying.

However, it can be seen from Table 2 that the parabolic model and the logarithmic model predicted the experimental results well with $R^2$ values greater than 0.98 and the lowest RMSE and $\chi^2$. Likewise, in the literature, the logarithmic model was successfully applied for pumpkin [24], while the parabolic model was suitable for Amasya red apples [25]. In addition, in a recent study working on the dehydration of pineapple, Izli et al. [10] found better fitting with the models of Two Term, Midilli et al., Page, and Wang and Singh (a particular version of the parabolic model).

### 3.2. Physical Quality of Pineapple

#### 3.2.1. Color

In every marketplace, consumers' preferences are essentially influenced by the surface color of food products. It is important to notice that food processing can preserve, facilitate, or damage the color of products; hence, color is a crucial index to consider. The appearance of different sample groups is laid out in Figure 4. The action of CD, RHCD, ID, and FD on the color parameters of pineapple slices is also displayed in Table 3. As a general remark, all drying methods profoundly modified the original color of fresh pineapples ($\Delta E > 3$). Among all these dehydration methods, FD recorded the highest brightness *(L\*)*, followed

by ID, while CD and RHCD slices showed no significant difference compared to fresh ones. *A** and *b** attributes incremented considerably (*p* < 0.05) after the drying processes. In contrast with FD, all heat-based operations exhibited the highest *a** and *b** values, with CD having the maxima in both parameters. The *a** parameter can allow us to foresee the degree of oxidation during the process. The highest *a** values at high temperatures were ascribed not only to enzymatic activities but, more importantly, to the caramelization of sugars in pineapples, which increased the redness of the samples [10]. However, ID had the lowest *a** value among thermally dehydrated slices, probably because of the short drying time.

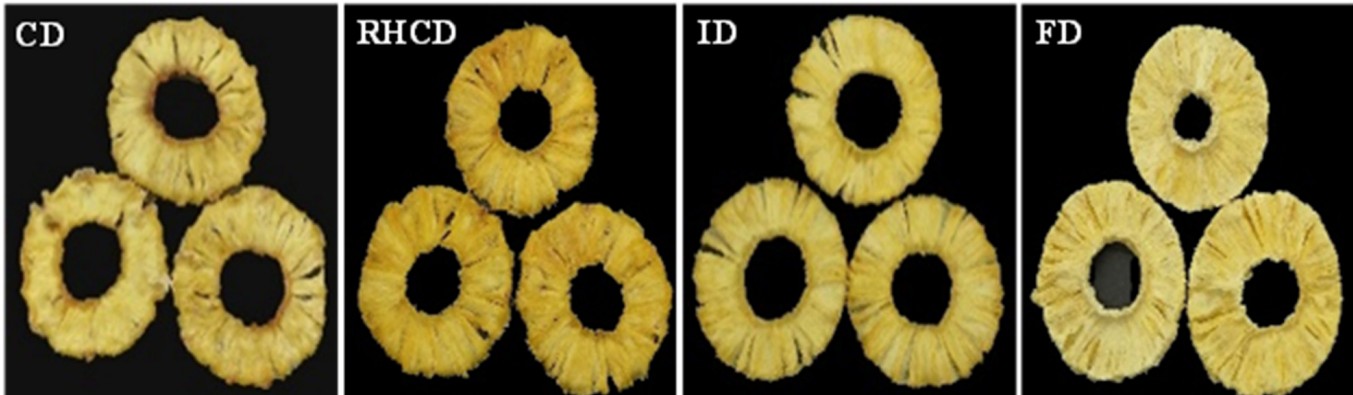

**Figure 4.** Digital images depicting the appearance of dried pineapple slices. Note: CD: convective drying, RHCD: relative humidity convective drying, ID: infrared drying, FD: freeze-drying.

**Table 3.** Color attributes of pineapple slices dried using different methods.

| Color Parameters | Fresh | CD | RHCD | ID | FD |
|:---:|:---:|:---:|:---:|:---:|:---:|
| *L** | 67.48 ± 1.20 [c] | 66.61 ± 1.55 [c] | 67.95 ± 1.42 [c] | 73.38 ± 1.70 [b] | 77.70 ± 1.33 [a] |
| *a** | −4.17 ± 0.63 [c] | −1.72 ± 0.54 [a] | −2.73 ± 0.33 [b] | −2.68 ± 0.50 [b] | −3.21 ± 0.69 [b] |
| *b** | 26.64 ± 0.91 [d] | 34.52 ± 0.85 [a] | 33.19 ± 0.94 [b] | 33.11 ± 1.61 [b] | 30.54 ± 1.17 [c] |
| ΔE | - | 8.30 ± 0.37 [c] | 6.72 ± 0.37 [d] | 8.88 ± 0.87 [b] | 10.98 ± 0.30 [a] |
| BI | - | 18.69 ± 0.71 [a] | 13.30 ± 0.87 [b] | 9.90 ± 0.77 [c] | 3.27 ± 0.18 [d] |

Note: Values are mean ± SD. Different letters in the same row show significant differences (*p* < 0.05). *a** is redness parameter, *b** is yellowness parameter, Δ*E* is color difference, *BI* is browning index; CD: convective drying, RHCD: relative humidity convective drying, ID: infrared drying, FD: freeze-drying.

The color difference of dried pineapples followed the pattern RHCD < CD < ID < FD. It is worth noting that although the color change was increased in ID and FD due to extremely high *L** values and reduced *a** and *b** (compared to other drying methods), ID and FD can lead to the improved acceptability of samples as consumers are more attracted to brighter colored and less-burned products [26].

To understand the degree of oxidation from each drying group, the browning index (*BI*) was calculated. This index was in line with the values obtained for the parameter *a** as the higher the *a** and *b** values, the greater the *BI*. Hence, CD had the largest *BI*, followed by RHCD, ID, and FD. Likewise, in a precedent study [27], hot air-dried hawthorn fruits had the highest *BI* compared to freeze-dried samples, and the outcome was due to higher *L** and lower *a** values in FD compared to oven-dried products.

Nevertheless, although the drying time of CD was slightly lower than that of RHCD, the *BI* was significantly lower in the latter. This observation is attributed to controlling the humidity in the RHCD dryer (20%), which contributed to limiting the oxidation of carbohydrates.

### 3.2.2. Texture

A texture profile analysis was performed to better evidence the action of dehydration on the texture of pineapples. A texture analyzer was used to imitate the perception during

mastication. It is clear from Table 4 that all investigated drying methods provoked some changes in textural parameters when compared to fresh materials. For example, RHCD, ID, and CD increased the hardness by 10.5, 9, and 7.8 times compared to the fresh group. Only FD values were similar to fresh food at $p > 0.05$. The increased hardness of dried fruits can be explained by eliminating water, which leads to the concentration of food compounds such as fibers and sugars.

**Table 4.** The texture of fresh and dried pineapple slices.

| Texture Parameters | Fresh | CD | RHCD | ID | FD |
|---|---|---|---|---|---|
| Hardness (g) | 236.25 ± 44.45 [c] | 1849.42 ± 323.13 [b] | 2488.27 ± 255.17 [a] | 2145.79 ± 208.01 [b] | 252.90 ± 35.19 [c] |
| Springiness (mm) | 0.94 ± 0.06 [a] | 0.88 ± 0.08 [ab] | 0.97 ± 0.05 [a] | 1.04 ± 0.24 [a] | 1.03 ± 0.22 [a] |
| Cohesiveness | 0.16 ± 0.06 [d] | 0.47 ± 0.16 [c] | 0.72 ± 0.10 [ab] | 0.69 ± 0.12 [b] | 0.81 ± 0.04 [a] |
| Gumminess | 33.22 ± 5.23 [e] | 530.96 ± 37.47 [c] | 1173.37 ± 190.45 [a] | 837.75 ± 161.15 [b] | 216.78 ± 32.98 [d] |
| Chewiness (mJ) | 31.42 ± 6.46 [e] | 411.36 ± 65.12 [b] | 851.11 ± 135.15 [a] | 138.20 ± 33.47 [d] | 267.31 ± 45.74 [c] |
| Resilience | 0.65 ± 0.01 [b] | 0.61 ± 0.13 [b] | 0.70 ± 0.10 [a] | 0.48 ± 0.09 [c] | 0.27 ± 0.02 [d] |

Note: values are shown as mean ± SD. Different letters in the same row show significant differences ($p < 0.05$). CD: convective drying, RHCD: relative humidity convective drying, ID: infrared drying, FD: freeze-drying.

In FD, the dehydration by sublimation resulted in the preserved porous structure and low shrinkage of samples compared to thermal treatments, thus ending up with softer products. This explanation was supported during the dehydration of cabbage under freeze-drying, hot air drying, vacuum drying, and microwave drying, and their combinations [28]. Additionally, because pineapples are rich in sugars, caramelization could have been responsible for the significant difference between thermal treatments and FD [29]. The springiness of samples, representing their elasticity, was practically not statistically modified after the dehydration processes. However, the springiness of CD was inferior to that of the fresh food. Cohesiveness figures were in the order FD < RHCD < ID < CD < fresh, meaning FD has the highest ability to maintain its shape between the two successive compressions. The gumminess, the energy needed to disintegrate a food, was the single parameter that showed the same trend as the hardness. It was found that the harder the pineapple sample, the higher the energy required for its disintegration. In general, different dehydration technologies increased the chewiness. It was noticed that RHCD, CD, and FD products were the chewiest. Resilience is the index that depicts the recovery of the food after compressions. Apart from RHCD, all drying groups exhibited a reduction in this parameter in contrast with the fresh pineapple. It is evidenced that heat-based procedures modified most of the texture parameters compared to FD. FD recorded the lowest hardness, gumminess, chewiness, and resilience, while RHCD had the greatest values in these respective attributes. This aligns with the research of Boateng and Yang [8], where the higher the processing temperature and time, the higher the textural modification after freeze-drying, infrared drying, convective drying, and pulsed-vacuum drying.

### 3.2.3. Rehydration and Microstructure

The rehydration test predicts the internal transformations that occur during processing, and most dried products are expected to be rehydrated before consumption. Pineapple slices were rehydrated for 30 min, and the results are presented in Figure 5. It can be seen that after immersion, FD samples absorbed moisture rapidly and had the highest rehydration ratio (RR). At the same time, RHCD registered the smallest RR, preceded by ID and CD. These variations in RR could be ascribed to cellular modifications during drying.

Images obtained from scanning electron microscopy were used to examine structural modifications triggered by different dehydration technologies (Figure 6). FD microstructures show numerous pores used as channels to incorporate water during the rehydration process. In contrast, thermally dried products had reduced RR compared to FD because of the excessive shrinkage of cells, which can be observed from the micrographs. However,

the shrinkage phenomenon was more pronounced in RHCD and CD than in ID; therefore, the rehydration ability was the highest for ID among them. In RHCD, the microstructure was particularly characterized by the pronounced destruction of the cellular structure. Consequently, there are fewer cellular organizations capable of retaining water during rehydration. This observation well justifies the lowest rehydration ratio obtained in relative humidity convective dried products. It can be concluded that the rehydration ratio is an excellent metric to judge the extent of internal modifications that occur in materials during drying.

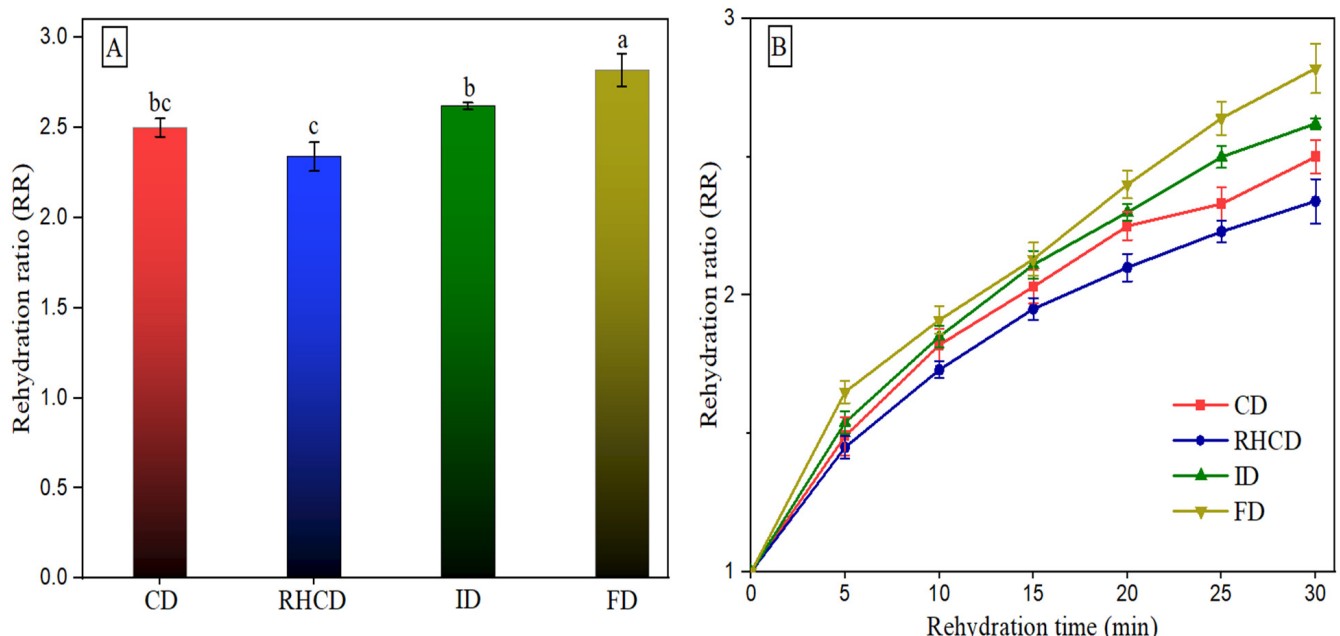

**Figure 5.** Rehydration of dried pineapple slices obtained after different drying techniques. (**A**) Rehydration ratio at the end of the rehydration process. Different letters (a, b, c) show statistical difference at $p < 0.05$ (**B**) Evolution of the rehydration ratio with soaking time. Note: CD: convective drying, RHCD: relative humidity convective drying, ID: infrared drying, FD: freeze-drying.

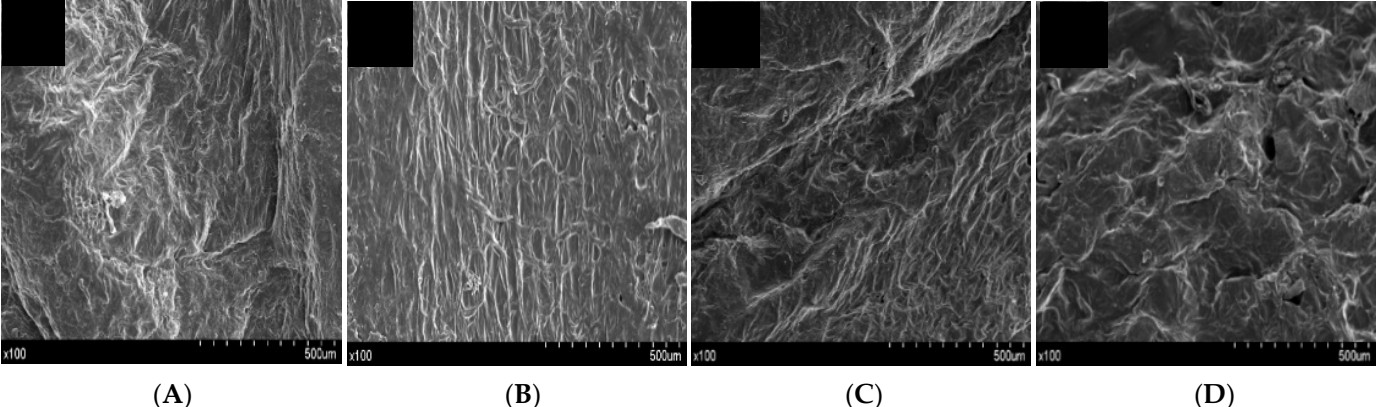

**Figure 6.** Microstructural images of dried pineapple slices ($\times$ 100) (**A**). Convective drying (CD), (**B**) relative humidity convective drying (RHCD), (**C**) infrared drying (ID), (**D**) freeze-drying (FD).

### 3.3. Aroma Profile Using Electronic Nose

The aroma of pineapple is one of the most important characteristics that influence its acceptability, and because of this sensory aspect, pineapple can enter various preparations as a flavoring agent. The electronic nose device was used to understand the aroma elements of dried products and their importance. Comparison was made only among dried

products since they have a similar moisture content and, therefore, a similar entrainment of volatile compounds.

Figure 7A is obtained by picking the maximum response given by each sensor. The major responses among all sensors were found in S7 (organic sulfides), S9 (aromatic ingredients, inorganic sulfides, and organic compounds), S6 (methyl groups), S2 (nitrogen oxides), and S8 (alcohols, ketones, and aldehydes), respectively. The remaining sensors (S1, S3, S4, S5, and S10) kept steady values regardless of the dehydration technique. According to the radar plot, RHCD, ID, and CD had the greatest volatile concentrations compared to FD, with RHCD and FD recording the highest responses. It can be deduced that the high temperature (60 °C) might have extracted and released more aromatic compounds than the operation at a lower temperature (FD). The same trend was reported in dried golden pompano fillets [30]. Lim et al. [31] stated that pineapple fruits do not lend themselves well to freezing as they tend to develop off flavors. This finding could explain the reduced flavor profile of freeze-dried pineapple slices. The results allow us to infer that different drying technologies induce specific patterns in the aroma profile of pineapples.

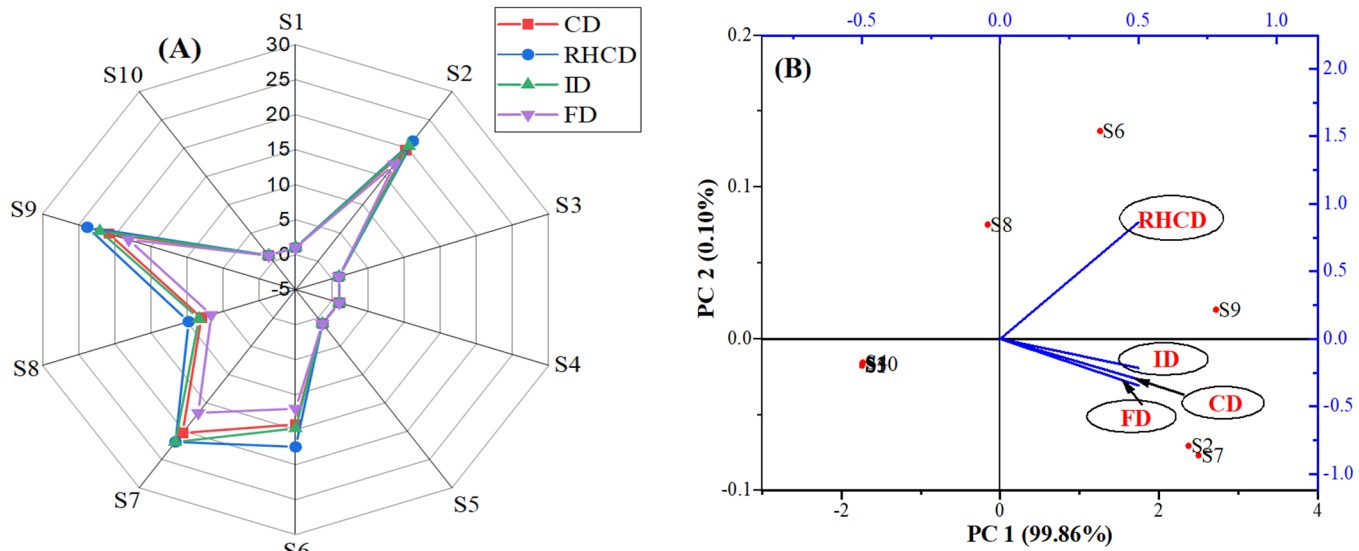

**Figure 7.** Effect of different drying techniques on the aroma of pineapple. (**A**) E-nose sensor responses for dried pineapple slices, (**B**) principal component analysis (PCA) of the aroma profile of different sample groups. Note: CD: convective drying, RHCD: relative humidity convective drying, ID: infrared drying, FD: freeze-drying.

To clearly differentiate the sample's results, the PCA analysis was performed (Figure 7B). Two groups of samples were identified. The RHCD sample was in the upper area, and other dried products were in the lower area. The PCA suggests a clear difference between these two sample groups. However, the aroma profiles of FD, CD, and ID were in a narrow range, with ID being superior followed by CD and FD. Therefore, the aroma of dried pineapple slices followed the order: RHCD > ID > CD > FD. Figure 7B also clarifies that compounds detected by sensors S6 and S9 were primarily responsible for the volatile results in RHCD, whereas S2 and S7 elements mostly influenced the aroma of ID, CD, and RHCD.

### 3.4. Enzymatic Activity

The enzyme activities (PPO and POD) of fresh and dried pineapples are presented in Table 5. To prolong the preservation of foods, it is crucial to inactivate oxidative enzymes whose action can reduce the shelf life and nutritional value of products. Fresh slices exhibited significantly higher enzyme activity than dried samples ($p < 0.05$). Drying inactivated oxidative enzymes resulted in PPO and POD activity reductions of 79.41% and

76.51% for CD, 88.24% and 67.63% for RHCD, 94.12% and 83.81% for ID, and 79.41% and 55.92% for FD, respectively. FD recorded the greatest activity figures compared to CD, RHCD, and ID. This observation proves the tremendous capacity of high temperatures to inactivate enzymes, while in FD, enzymes were kept less operative during the process due to lower temperatures but were active when the conditions became favorable. This enzymatic trend was found in Osae et al. [7] where ginger morsels dried under microwave drying, freeze-drying, infrared drying, and relative humidity convective drying exhibited enzyme inactivation rates of 10–79% for PPO and 18–83% for POD, with the freeze-drying process having the lowest inactivation level.

**Table 5.** Enzyme activities of fresh and dried pineapple slices under various processing conditions.

| Enzyme Activities (U/g) | Fresh | CD | RHCD | ID | FD |
|---|---|---|---|---|---|
| PPO | 0.34 ± 0.02 [a] | 0.07 ± 0.02 [b] | 0.04 ± 0.01 [bc] | 0.02 ± 0.00 [c] | 0.07 ± 0.02 [b] |
| POD | 31.88 ± 1.38 [a] | 7.49 ± 0.47 [d] | 10.32 ± 1.23 [c] | 5.16 ± 0.93 [e] | 14.05 ± 1.65 [b] |

Note: Values are mean ± SD. Different letters in the same row point out statistical differences ($p < 0.05$). PPO: polyphenol oxidase, POD: peroxidase, CD: convective drying, RHCD: relative humidity convective drying, ID: infrared drying, FD: freeze-drying.

Among thermal dehydration techniques, ID had the lowest enzymatic activity. The effect of infrared radiations provoked the homogeneous heating of the pieces, resulting in the increased inactivation of PPO and POD. The activity of peroxidase in the fresh and after-drying samples was the highest compared to that of PPO which must gather much attention in the processing of pineapple as POD activity represented 94 times that of PPO in the fresh sample. Overall, enzyme inactivation followed the order ID > CD > RHCD > FD. It has also been demonstrated that infrared radiation can be used not only for drying crops but also for deactivating enzymes [32].

## 4. Conclusions

Applying infrared drying, convective drying, relative humidity, convective drying, and freeze-drying will act as a driving force for the further development of the pineapple operations systems. Moreover, the effects of various drying methods on the drying kinetics, physical quality, aroma, and enzymatic activity of pineapple slices were studied. The $D_{eff}$ ranged from $5.89 \times 10^{-9}$ m$^2$/s to $2.32 \times 10^{-8}$ m$^2$/s, while the parabolic model and the logarithmic model were the best mathematical models that well predicted the kinetics of the moisture ratio during all investigated drying techniques. FD recorded satisfactory quality attributes such as the greatest rehydration ratio, minor microstructural modifications, and the lowest browning index. However, it was time intensive and resulted in products having lower aroma concentrations and higher enzyme activities. Therefore, a longer processing time during FD may lead to increased energy consumption, processing costs, and the price of finished products. According to our findings, infrared drying is the optimal processing technology for dehydrating pineapple, considering drying time and different quality parameters. The ID had the lowest drying time with brighter color and reduced browning index, high rehydration capacity, excellent aroma profile, and a great ability to inactivate oxidative enzymes. Therefore, from an industrial viewpoint, infrared drying could permit the obtaining of highly attractive pineapple slices and reduce drying time and, consequently, the energy required to complete the drying process and the cost of end products. The above-mentioned effects of ID will, therefore, contribute to solving the issue of fruit degradation and increasing the marketableness of dried products. Future studies could employ more robust techniques such as gas chromatography or sensory analysis to align the results with the organoleptic perceptions of a panel of evaluators to supplement the E-nose results. Moreover, future investigations will focus on preserving nutrients and applying various pretreatments for process improvement on both the quality and drying kinetics of pineapple.

**Author Contributions:** Conceptualization, E.S.T. and B.X.; Methodology, E.S.T.; Software, E.S.T. and I.D.B.; Validation, I.D.B.; Formal analysis, E.S.T.; Investigation, E.S.T.; Resources, C.Z. and B.X.; Data curation, E.S.T.; Writing—original draft, E.S.T.; Writing—review & editing, I.D.B., C.Z. and B.X.; Funding acquisition, B.X. All authors have read and agreed to the published version of the manuscript.

**Funding:** This research was funded by the National Key Research and Development Program of China (2022YFF11001700).

**Data Availability Statement:** Data are contained within the article.

**Conflicts of Interest:** The authors declare that they have no known competing financial interests or personal relationships that could have appeared to influence the work reported in this paper.

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
