# Peer review of "Role of Drying Technologies on the Drying Kinetics, Physical Quality, Aroma, and Enzymatic Activity of Pineapple Slices"

_sustainability, doi:10.3390/su152215906_

Round 1

Reviewer 1 Report

Comments and Suggestions for Authors

General comment:

The text is written and edited carelessly. No capital letters in chapter titles, poor paragraph alignment, varied font. The document does not include line numbering making it very difficult to pinpoint what specifically needs to be improved.

Abstract:

The abstract is written in two different fonts and is far too long. It contains unnecessarily too many results such as values of determination coefficients. Please shorten the abstract, standardize the font.

Introduction

In the section on the drying techniques used, the authors should also refer to reports involving fruits other than pineapple.

Material and results

“2.2 Drying procedures” should be on a new line.

The numbering of mathematical formulas is non-uniform. Equation 3 has no reference.

2.4.1. color” – section title should be capitalized.

Under equation number 15 there is another formula which is not in the right place because it is on the text.

2.4.4. microstructure” - section title should be capitalized.

Tables 3 and 5 needs to be revised as it is unreadable in its current state

Conclusions

“The dehydration techniques influenced in a specific manner the investigated parameters as each dryer is based on a particular operation principle.” - This sentence presents the obvious, if the drying methods were not different the whole work would not make sense.

The summary should include a specific statement as to the optimal method from the point of view of various quality characteristics.

Comments on the Quality of English Language

The document should be checked for correct language.

Author Response

attached is reviewer 1 response

Reviewer 2 Report

Comments and Suggestions for Authors

The actual objective of this work is somewhat unclear, especially considering that the industry commonly prefers freeze-drying for its effectiveness in preserving sensory quality and bioactive compound composition. Given that the conclusion suggests that traditional drying (ID) might be superior, it's essential to consider the potential industrial applications arising from these results.

Add a sub-unit in the methodology section for Convective Drying.

Consider moving the content from the first column in Tables 3 and 5 to a paragraph within the discussion section, as it appears to be more suited for that context.

Place Table 4 within the Material and Methods section.

Include a dedicated section related to statistical analyses in the Material and Methods.

The PCA obtained through electronic nose results can be somewhat deceptive, as fresh products, with their higher water content, also have a higher vapor pressure. This leads to the natural entrainment of volatile compounds, making higher-moisture products more aromatic. In contrast, dehydrated products, with lower moisture and vapor pressure, won't have the same entrainment of volatile compounds and may be perceived as "less aromatic." However, it is known, for instance, that freeze-drying retains volatile compounds that contribute to aroma and can be perceived in the nose and mouth, forming what we call "flavor."

To corroborate this, it could be useful to employ more robust techniques such as gas chromatography or conduct sensory analysis to align the results with the organoleptic perceptions of a panel of evaluators.

Comments on the Quality of English Language

Moderate English language editing can be performed

Author Response

Attached is reviewer 2 response

Reviewer 3 Report

Comments and Suggestions for Authors

The main purpose of this research was to investigate the effects of four distinct drying methods on pineapple slice dehydration, with a focus on their influence on drying rates, physical characteristics, aroma, and enzymatic activity, as documented in the study.

In general, this manuscript presented relevant results about the effect of different drying methods of pineapple slices on their quality properties.

There are some specific comments listed below:

Tables 3 and 5 are unconfigured.

Some properties such as rehydration rate, microstructure, etc. did not indicate their respective replications.

The references are cited appropriately.

The results obtained by aroma are not presented clearly. What do the results obtained in the different drying processes mean? What is the purpose of comparing with the fresh product?

The authors stated in the conclusion that the ID drying process presented an "excellent flavor profile". What result proves this statement? In my opinion, this aroma result does not add any relevant information about the drying processes of pineapple slices.

Author Response

attached is reviewer 3 response

Round 2

Reviewer 1 Report

Comments and Suggestions for Authors

The manuscript has been significantly improved and, in my opinion, should be accepted for publication.

Reviewer 2 Report

Comments and Suggestions for Authors

Adjustments have been made accordingly throughout the document. It can be published now.

Comments on the Quality of English Language

Minor aspects can be adjusted prior to publication 

Reviewer 3 Report

Comments and Suggestions for Authors

Dear authors,

Thank you for the answers to the questions asked by this advisor.

In my opinion, the article is ready to be published.

Congratulations.